# Silver Nanoparticle/Carbon Nanotube Hybrid Nanocomposites: One-Step Green Synthesis, Properties, and Applications

**DOI:** 10.3390/nano13081297

**Published:** 2023-04-07

**Authors:** Jun Natsuki, Toshiaki Natsuki

**Affiliations:** 1Institute for Fiber Engineering (IFES), Interdisciplinary Cluster for Cutting Edge Research (ICCER), Shinshu University, 3-15-1 Tokida, Ueda, Nagano 386 8567, Japan; jnatsu@shinshu-u.ac.jp; 2Faculty of Textile Science and Technology, Shinshu University, 3 15 1 Tokida, Ueda shi, Nagano 386 8567, Japan

**Keywords:** silver nanoparticles, carbon nanotubes, synthesis, green one-step method

## Abstract

Hybrid nanocomposites of silver nanoparticles and multiwalled carbon nanotubes (AgNPs/MWCNTs) were successfully synthesized by a green one-step method without using any organic solvent. The synthesis and attachment of AgNPs onto the surface of MWCNTs were performed simultaneously by chemical reduction. In addition to their synthesis, the sintering of AgNPs/MWCNTs can be carried out at room temperature. The proposed fabrication process is rapid, cost efficient, and ecofriendly compared with multistep conventional approaches. The prepared AgNPs/MWCNTs were characterized using transmission electron microscopy (TEM) and X-ray photoelectron spectroscopy (XPS). The transmittance and electrical properties of the transparent conductive films (TCF_Ag/CNT) fabricated using the prepared AgNPs/MWCNTs were characterized. The results showed that the TCF_Ag/CNT film has excellent properties, such as high flexible strength, good high transparency, and high conductivity, and could therefore be an effective substitute for conventional indium tin oxide (ITO) films with poor flexibility.

## 1. Introduction

High electrical conductivity and high optical transparency are desired for transparent conductive films (TCFs). Commonly used TCFs, such as indium tin oxide (ITO) film, have been widely applied to touch screen, panel display, liquid crystal displays, and solar cells [1,2,3]. However, ITO films are high cost owing to the lack of indium source. In addition, the resistance of ITO thin films increases remarkably with bending force. The brittleness of ITO has limited its applications in flexible electronic devices owing to the low bending strength and particle size of indium [4,5]. Thus, ITO films must be replaced with cost effective and high-performance flexible materials. **Alternative materials, such as carbon nanotubes** (CNTs) decorated with silver nanoparticles (AgNPs), have been studied for their synthesis **and electricity** evaluation. CNTs, which are cylinder-shaped allotropic forms of carbon, exhibit interesting mechanical and physical properties, such as high strength, unique electronic state, and efficient heat conductivity and optical transmission [6,7,8,9]. CNTs have novel properties that make them suitable in many applications, including engineering, nanotechnology, and materials science [10,11,12,13].

The **synthesis** of AgNPs has received remarkable attention in the field of materials science [14,15,16,17]. AgNPs are widely used in the electronics industry for printed electronic circuits due to their advantages, such as high electrical conductivity and nanometer scale [18,19,20,21]. However, the fragility of AgNPs results in the lack of flexibility of the film prepared from this material. Given the individual benefits of CNTs and AgNPs, their composites can reach highly effective materials that cannot be achieved independently [22].

Many works have described fabrication processes for flexible TCFs using CNTs [23,24,25,26,27,28,29]. Transparent CNT film production mainly includes drop-drying from solvent, airbrushing, and Langmuir–Blodgett deposition. An early work by Geng et al. [24] reported a relatively straightforward approach to fabricate large-sized TCFs using single-walled CNTs (SWCNTs) by spray coating using sodium dodecyl sulfate (SDS)-dispersed SWCNTs. A comprehensive review was conducted on CNT TCFs and detailed fabrication methods, such as film patterning, chemical doping effects, and hybridization with other materials [25]. The review also focused on the optoelectronic properties of the films and their potential applications of photovoltaics, touch panels, and liquid crystal displays. Zhou and Azumi [27] reviewed and summarized the recent progress in the fabrication, properties, and application of CNT-based TCFs to open new opportunities in the development of next-generation flexible displays. They also gave some possible strategies to reduce the production cost and improve the conductivity and transparency. Hou et al. [28] obtained high-quality double-walled CNT-based TCFs fabricated by scalable filtration and showing extremely low sheet resistance of 83 Ω/sq with transmittance of 79% at 550 nm.

AgNPs are well known for their high electrical conductivity and antimicrobial properties. Adding AgNPs can effectively improve the electrical conductivity of nanocomposites and provide good antibacterial properties compared with pristine CNTs. Some studies were conducted on various CNT films decorated with various AgNP concentrations [30,31,32,33,34,35]. Yu et al. [31] used a potentiostatic double-pulse technique to electrodeposit AgNPs onto multiwalled CNT (MWCNTs) multilayer films preassembled on ITO. The results showed that AgNPs of 10–500 nm with varied density could be electrogenerated on MWCNT surface, and the MWCNT electrodes exhibited good conductivity. Lin et al. [32] developed a novel nonenzymatic sensor with high sensitivity using polydopamine (PDA) as a multifunctional intermediate for depositing AgNPs onto SWCNTs. The advantages of the sensor could be ascribed to using CNTs and AgNPs. Due to the large surface area, CNTs were used as an effective matrix for AgNP loading due to the large surface area to improve the conductivity of the nanocomposite, and AgNPs were attached to SWCNTs through in situ chemical reduction by PDA to improve the surface active for electrochemical catalysis. Shi et al. [35] reported that cetylpyridinium borohydride (CBH4)-coated MWCNT could be used as a template for the synthesis of AgNPs on MWCNTs. The obtained AgNP/MWCNT hybrid material showed a promising electrocatalytic performance. Although these studies explored synthesis procedures for hybrid nanostructures consisting of CNTs and AgNPs using chemical reaction methods, only a few reports focused on CNT-based nanocomposites decorated with AgNPs for preparing high-performance TCFs [36,37,38]. Lee et al. [36] prepared TCFs fabricated with DWNT-AgNP suspension using a wet coating method. The prepared DWNT-AgNP thin films had a low sheet resistance of 53.4 Ω/sq and an optical transmittance of 90.5%. Ko et al. [37] showed that the sheet resistance remained approximately 370 Ω/sq, even after 500 bending cycles, with a transmittance of around 77%. Li et al. [38] achieved optoelectronic properties with sheet resistance of 50.3 Ω/sq and a high transmittance of 79.73% at a wavelength of 550 nm.

In this work, we proposed a one-step green method to decorate AgNPs on the surface of MWCNTs. Organic solvent was not used, and the reaction proceeded at room temperature. AgNP/MWCNT nanocomposites with uniform, 5 nm diameter AgNPs on the surface of MWCNTs were effectively obtained by simple separation. The developed method is suitable for practical applications because it is an easy and environmentally benign process. Furthermore, electronic circuits prepared with the AgNP/MWCNT nanocomposites can be manufactured easily by sintering at low temperature because no dispersing polymer was used. The transmittance and electrical properties of the TCF_Ag/CNT film fabricated using the AgNP/MWCNT nanocomposites were also characterized. The results showed that the TCF_Ag/CNT film has excellent properties, such as high flexible strength, good transparency, and high conductivity.

## 2. Experimental Section 

### 2.1. Materials

Sodium gluconate, sodium citrate (Na_3_Ct), silver nitrate (AgNO_3_), and dimethylaminoethanol (DMAE; 2-dimethylaminoethanol) were purchased from Wako Inc., Japan. Nitric acid (HNO_3_) and lysine (L(+)-Lysine) (Wako, Inc., Tokyo, Japan) were used as solvents. Gallic acid (gallic acid monohydrate) was obtained from Wako, Japan. Deionized water was used throughout the experiments.

### 2.2. Preparation of Functionalized MWCNTs

MWCNTs (d = 20–30 nm, Wako Inc., Tokyo, Japan) were used in this study. CNTs have high physical durability, lightweight, flexibility, and excellent electrical and thermal conductivity. CNTs exhibit low reactivity because they lack functional groups. For easy dispersion and bonding of AgNPs onto the CNT surfaces, the CNTs were modified to introduce functional groups onto their surfaces. The functional group introduced onto the CNTs was not particularly limited, forming a hydrogen bond (intermolecular force bond) with a hydroxyl group (-OH) or a carboxylic acid (-COOH) group. In this work, two kinds of methods using nitric acid (HNO_3_) and lysine (L(+)-Lysine) were used to modify the MWCNTs.

Nitric acid-modified MWCNTs were obtained through the following steps. First, 0.15 g of MWCNTs and 70 mL of HNO_3_ were placed in a flask and stirred at 120 °C for 10 h in an oil bath. The obtained reaction solution was then centrifuged at 5000× *g* rpm for 1 min, and the supernatant was discarded. The precipitate was washed with pure water and centrifuged twice at 5000 rpm for 2 min. The modified CNTs were placed in 1 L of pure water and left for 24 h to neutralize their pH. After centrifugation at 5000 rpm for 1 min, the supernatant was discarded, and the residue was dried at 60 °C for 24 h to obtain nitric acid-modified MWCNTs.

Lysine-modified MWCNTs were obtained as follows. In brief, 0.25 g of lysine was stirred in a beaker with 120 mL of purified water at 60 °C until completely dissolved. The solution was then added with 0.25 g of MWCNTs, stirred at 60 °C for 2 h, and centrifuged at 5000 rpm for 1 min. The supernatant was discarded, and the precipitate was washed with pure water and centrifuged twice at 5000× *g* rpm for 2 min. The modified CNTs were placed in 1 L of pure water and left for 24 h to neutralize their pH. After centrifugation at 5000 rpm for 1 min, the supernatant was discarded and dried at 60 °C for 24 h to obtain lysine-modified MWCNTs.

When nitric acid is used for the surface treatment of MWCNTs, the procedure must be conducted at high temperature for a long time. The method also requires time and effort to make the sample neutral because a strong acid is used. Therefore, we developed a simple treatment method that uses lysine and has the merit of processing at a low temperature and in a short period of time.

### 2.3. Fabrication of AgNP/MWCNT Nanocomposites

A simple, green one-step synthesis method was developed to synthesize AgNP/MWCNT nanocomposites. Figure 1 shows the fabrication of AgNPs/MWCNTs in which MWCNTs were used as a dispersing agent. All the above steps were performed at room temperature.

In brief, 0.04 g of the modified MWCNTs (Section 2.2) were added to 20 mL of pure water and stirred at room temperature for 10 min. The solution was then added with 0.5 g of silver nitrate and stirred for another 10 min, followed by the addition of an aqueous solution of gallic acid dissolved in 30 mL of pure water and 0.262 g of DMAE, stirring for 1 h. The obtained reaction solution was centrifuged at 5000 rpm for 1 min, the supernatant was discarded, and the precipitate was washed with pure water. After the steps were repeated twice (centrifugation at 5000 rpm for 2 min), the precipitate was redispersed in 10 mL of ethanol to obtain an aqueous dispersion of AgNP/MWCNT nanocomposites. For the preparation of pure AgNPs/MWCNTs, the resulting product was centrifuged, washed with pure water, and dried overnight at 60 °C.

### 2.4. Preparation of AgNP/MWCNT Film (TCF_Ag/CNT)

**Polyethylene** terephthalate (PET, U34, 100 μm, Co., Ltd., Toyama, Japan) film was used as the substrate of the TCFs. The PET films were cleaned using an ultrasonic cleaner **with ethanol** for about 10 s to remove surface debris, washed with pure water, and dried. TCF_Ag/CNT was prepared by coating the PET surface with AgNPs/MWCNTs.

The coating method was as follows: AgNPs/MWCNTs were dispersed in isopropyl alcohol (IPA, Wako Inc., Tokyo, Japan). The PET substrate was placed on a hot plate at a temperature of 60 °C. With the spray method, TCF_Ag/CNT was coated at 15 cm distance between the substrate and spray nozzle as shown in Figure 2. TCF_Ag/CNT was fabricated using two types of modified CNT: nitric acid-modified (CNT_HNO_3_) and lysine-modified (CNT_lyine). TCF_Ag/CNT films were prepared using 0.001, 0.005, and 0.01 g of AgNPs/MWCNTs under nitric acid modification and denoted as TCF_Ag/CNT_HNO_3_ 0.001 g, TCF_ Ag/CNT_ HNO_3_ 0.005 g, and TCF_Ag/CNT_HNO_3_ 0.01 g, respectively. With the same method, TCF_Ag/CNT_lysine 0.001 g, TCF_Ag/CNT_lysine 0.005 g, and TCF_Ag/CNT_lysine 0.01 g were fabricated using the lysine-modified AgNPs/MWCNTs. The thickness of coating layer affects the light transmittance. The transmittance decreases with increasing film thickness. In order to obtain transmittance, the film thicknesses of all the samples are less than 1 μm in this work.

### 2.5. Characterization Methods

The AgNP/MWCNT nanocomposites were characterized using various measurements including UV–Vis spectroscopy (Hitachi U-4100 UV–Vis Spectrophotometer), X-ray diffraction (XRD) (Rigaku D/MAX-IIIV X-ray Diffractometer with Cu-Kα radiation), transmission electron microscopy (TEM, JEOL JEM2010 operating at 200 kV), energy dispersive X-ray spectroscopy (EDS) (FE-SEM, Hitachi S-5000 equipped with an EDS instrument), and X-ray photoelectron spectroscopy (XPS) (Kratos Axis Ultra DLD, Kratos Analytical Ltd., Manchester, UK). The electrical properties of samples were measured using a resistivity meter (Loresta-GP, MCP-T700, Mitsubishi Chemical Ltd., Tokyo, Japan).

## 3. Results and Discussion

### 3.1. Characterization of AgNPs/MWCNTs

TEM and XPS analyses were used to characterize the functional groups on the surfaces of MWCNTs and AgNP/MWCNT nanocomposites. Figure 3 and Figure 4 show the TEM images of AgNPs/MWCNTs nanocomposites. The AgNPs had a spherical shape and uniform diameter of about 5 nm and were homogeneously dispersed without agglomeration and strongly adhered to the surface of the MWCNTs. As shown in Figure 4, it was observed that the AgNPs are distributed uniformly on the CNT’s surface with high ratio in the nanocomposites, having a diameter of 20–30 nm. This phenomenon was attributed to the condensation reaction between the functional groups on the CNT surfaces and AgNPs, resulting in the AgNPs adhering to the CNTs without aggregation. The reduced AgNPs are covered with a reducing agent layer. Therefore, the AgNPs become very stable due to the strong binding of AgNPs with the functional groups introduced to the CNT surface through the covering layer.

Figure 5 shows the XPS survey spectra of modified MWCNTs, Ag/CNT_HNO_3_, and Ag/CNT_lysine. The peak of C1s, Ag3d, and Ag3p confirmed the existence of carbon and the formation of AgNPs. C1s showed a strong peak at 284 eV and O1s at 532 eV. The Ag3p peaks composed of Ag3p3/2 and Ag3p1/2 contained two strong peaks at 573 and 604 eV. Furthermore, the amplified region around the Ag3d peaks is shown in Figure 6. The XPS survey spectrum of Ag3d was composed of Ag3d5/2 and Ag3d3/2, showing strong peaks at 368 and 374 eV, respectively. The difference between the Ag3d5/2 and Ag3d3/2 peaks was 6 eV, which proved the formation of crystalline AgNPs.

### 3.2. Characterization of AgNP/MWCNT Films (TCF_Ag/CNT)

FE-SEM images and EDS analysis of TCF_Ag/CNT_HNO_3_ and TCF_Ag/CNT_lysine are shown in Figure 7. The AgNPs adhered to the TCF’s surface. The AgNP/MWCNT films contained AgNPs with high ratio and no impurities were found, as shown in the EDS analysis of Table 1.

### 3.3. Transmittance of AgNP/MWCNT Films

The relationship between the transmittance and wavelength waveform was measured for the TCF_Ag/CNT films prepared in Section 2.4. The transmittance at the visible wavelength of 550 nm was taken as the transmittance of the film. In this study, the transmittance of the substrate PET film is 100%, and the measured material is regarded as transparent when the transmittance of substrates exceeds 80%.

Figure 8 shows the variation of transmittance with the wavelength of TCF_Ag/CNT thin films with two different surface treatment methods. The TCF_Ag/CNT thin films fabricated using 0.01 g of AgNPs/MWCNTs had transmittances of 20.4% and 2.0% in the wavelength of 550 nm for TCF_Ag/CNT_lysine and TCF_Ag/CNT_HNO_3_, respectively. The transmittance of TCF_Ag/CNT_lysine was 10 times as high as that of Ag/CNT_HNO_3_. Figure 9 shows the relationship between the transmittance of TCF_Ag/CNT_lysine thin films and the wavelength under different amounts of added AgNPs/MWCNTs. The transmittance of TCF_Ag/CNT_lysine reached a high value of 89.7% in the wavelength of 550 nm when the amount of added AgNPs/MWCNTs was 0.001 g. The transmittance under the addition of 0.01 and 0.005 g of TCF_Ag/CNT_lysine was 59.1% and 20.4%, respectively. Therefore, films with high transmittance can be fabricated using AgNP/CNT_lysine nanocomposites.

### 3.4. Electrical Conductivity of AgNP/MWCNT Films

The electrical conductivity properties of the TCF_Ag/CNT films prepared in Section 2.4 were measured using Loresta-GP MCP-T610 resistivity meter (Mitsubishi Chemical Analytech Co., Ltd., Tokyo, Japan). Figure 10 shows the relationship between surface resistivity and transmittance for TCF_Ag/CNT_lysine and TCF_Ag/CNT_HNO_3_. The electric resistance of TCF_Ag/CNT_lysine was lower than that of TCF_Ag/CNT_HNO_3_, indicating that TCF can be successfully produced by the spray coating method. Moreover, we investigated the change in the electrical resistance of TCF_Ag/CNT_lysine before and after bending. The electrical resistance before bending was 22 × 10^4^ Ω, and the electrical resistance after bending was 35 × 10^4^ Ω. Almost no change of electrical resistance was observed before and after bending. The reason is that the MWCNTs have a high aspect ratio and superior mechanical and electrical properties. ITO is a good transparent conductive material applied to electronic devices. However, the ITO film has low flexible strength because of the sintered ITO particles with large size. In this study, the TCF film fabricated with AgNP/MWCNT nanocomposites is strong against bending due to the flexibility of CNTs. The TCF_Ag/CNT films have high mechanical flexibility, high transparency, and high conductivity, showing potential application to screen panels, thin film transistors, and field emission devices.

## 4. Conclusions

In this study, a simple, green one-step method was developed to synthesize AgNP/MWCNT nanocomposites. The synthesis and attachment of AgNPs onto the surface of MWCNTs were carried out simultaneously at room temperature without using any organic solvent. The AgNP/MWCNT nanocomposites are easily separated from solution because no organic solvent is used. Moreover, the TCF film fabricated with the AgNP/MWCNT nanocomposites can be sintered easily at a low temperature because no dispersing polymer was used in the fabrication. Taking advantage of the excellent properties of AgNPs and CNTs, the prepared TCF_Ag/CNT films have excellent properties, such as high flexible strength, good transparency, and high conductivity. These advantages render the TCF_Ag/CNT films an effective substitute for conventional ITO films with poor flexibility. Therefore, the developed method is suitable for practical applications because the manufacturing process has the advantage of environmental friendliness.

## Figures and Tables

**Figure 1 nanomaterials-13-01297-f001:**
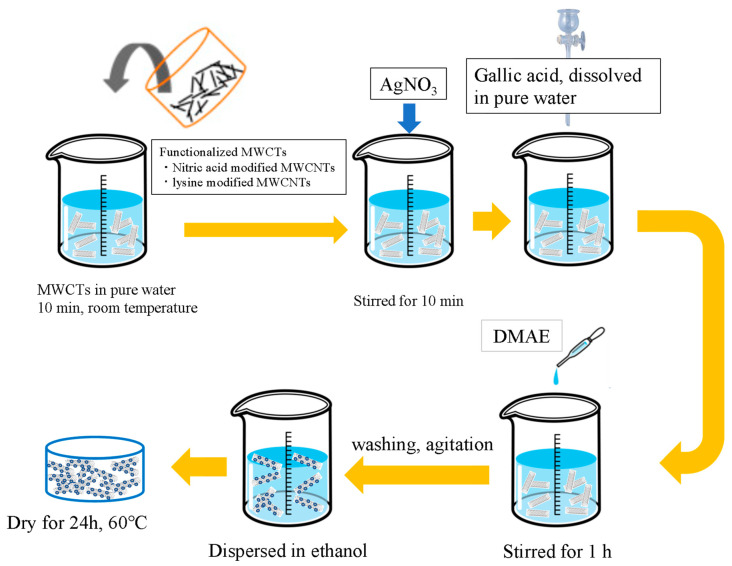
Fabrication process of AgNP/MWCNT nanocomposites.

**Figure 2 nanomaterials-13-01297-f002:**
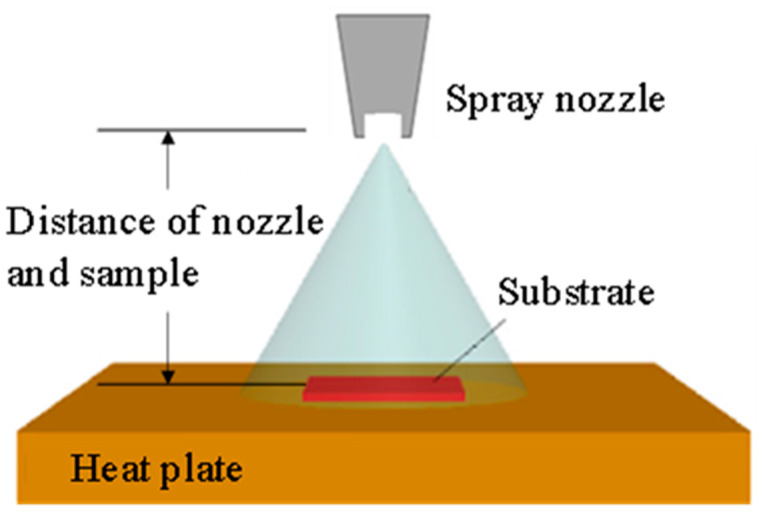
Schematic of spray coating deposition for thin film.

**Figure 3 nanomaterials-13-01297-f003:**
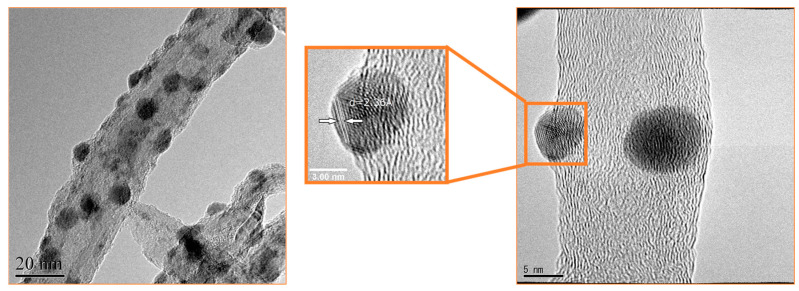
TEM micrographs of AgNP/MWCNT nanocomposites.

**Figure 4 nanomaterials-13-01297-f004:**
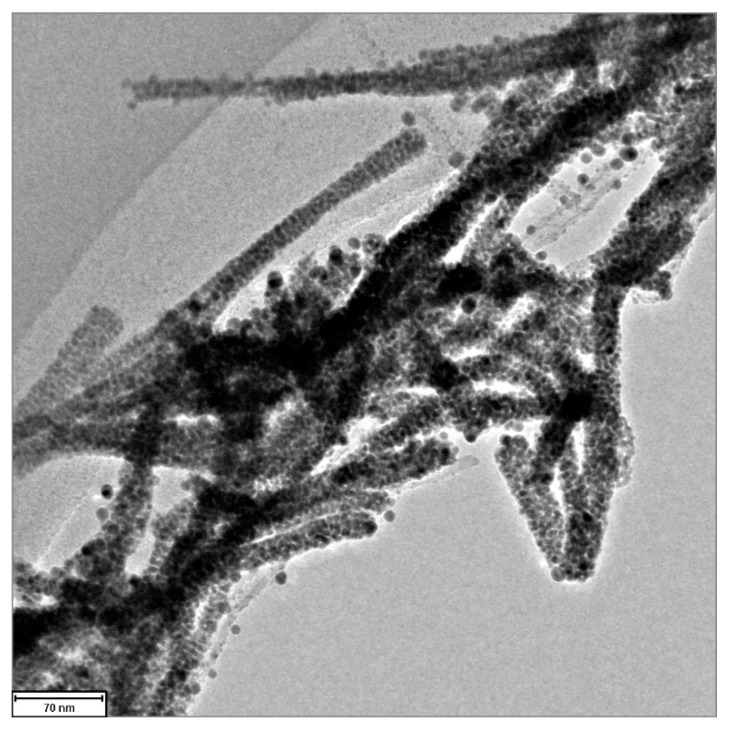
TEM micrograph of the distribution image of AgNPs attached on the CNTs.

**Figure 5 nanomaterials-13-01297-f005:**
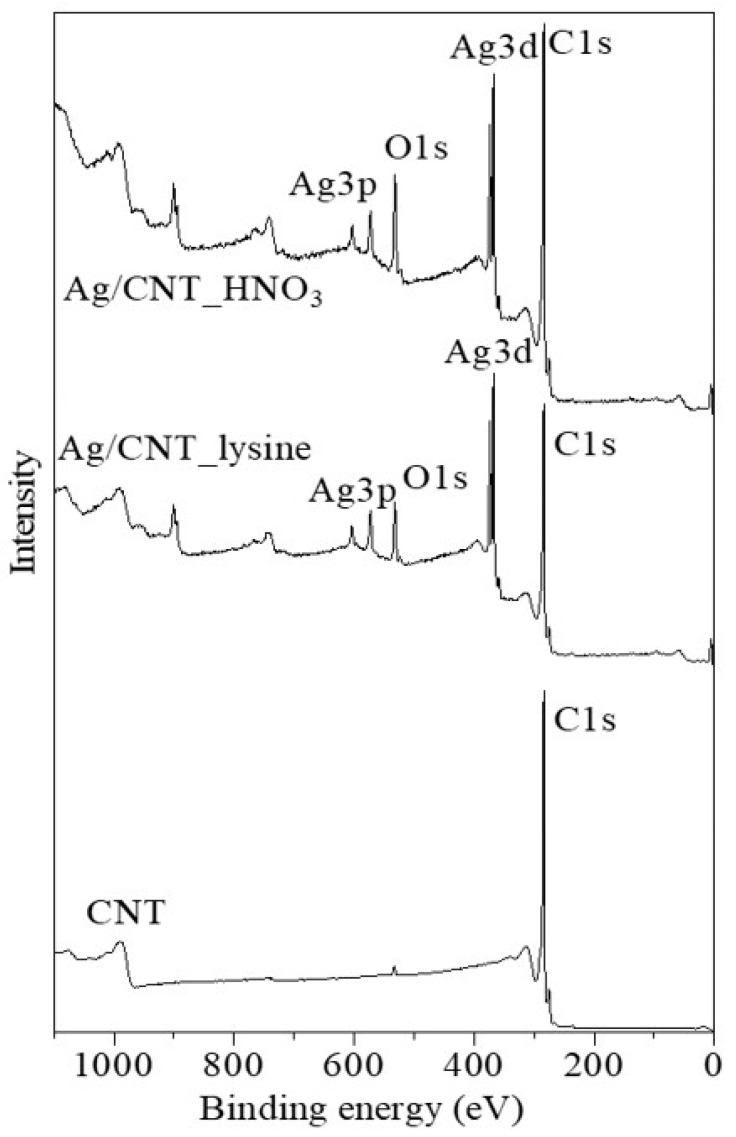
XPS survey spectrum of the AgNP/MWCNT nanocomposites.

**Figure 6 nanomaterials-13-01297-f006:**
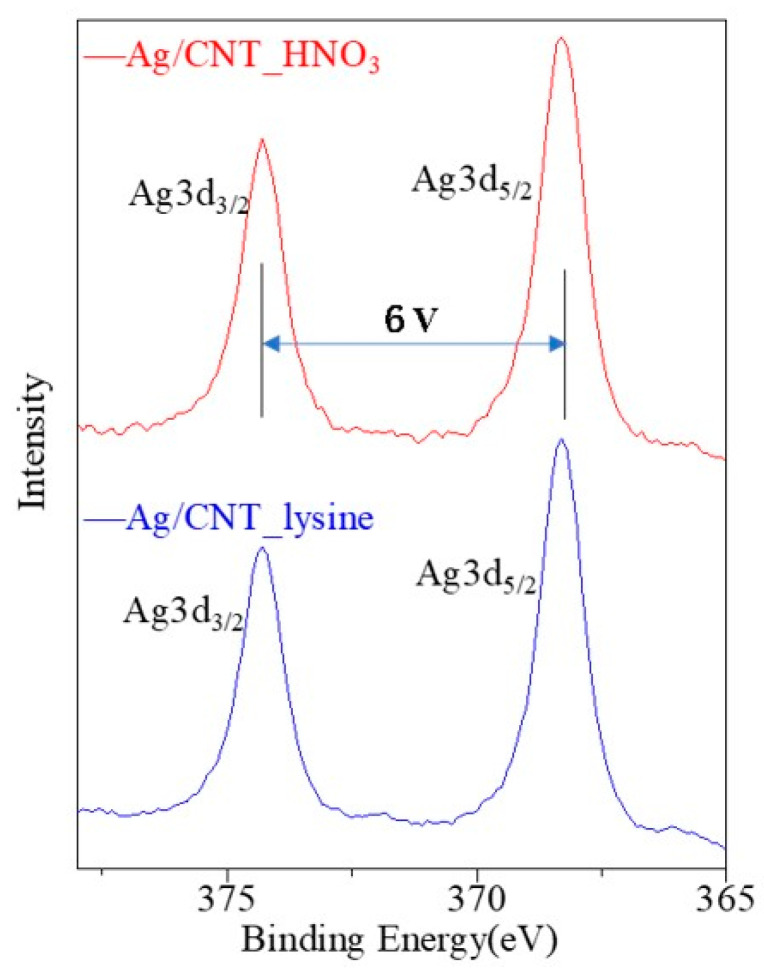
High-resolution XPS survey spectrum of the Ag3d peak.

**Figure 7 nanomaterials-13-01297-f007:**
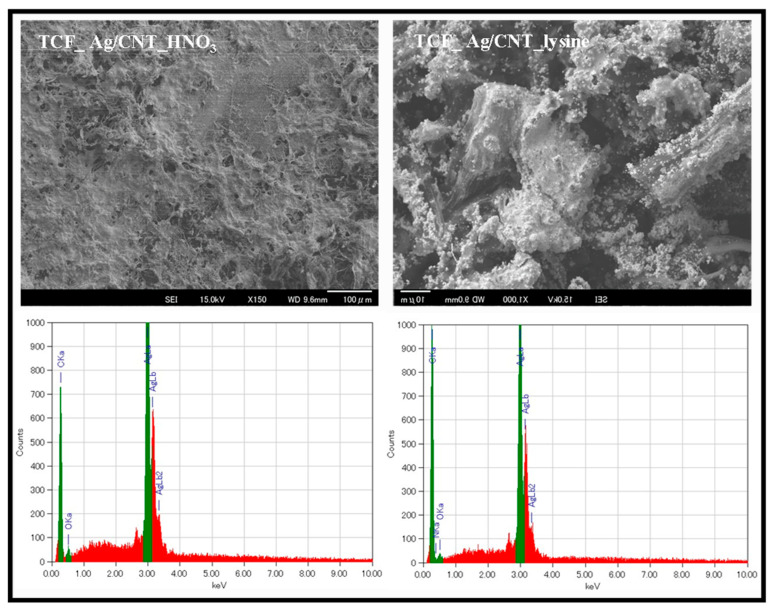
FE-SEM images and EDS analysis of AgNP/MWCNT nanocomposites.

**Figure 8 nanomaterials-13-01297-f008:**
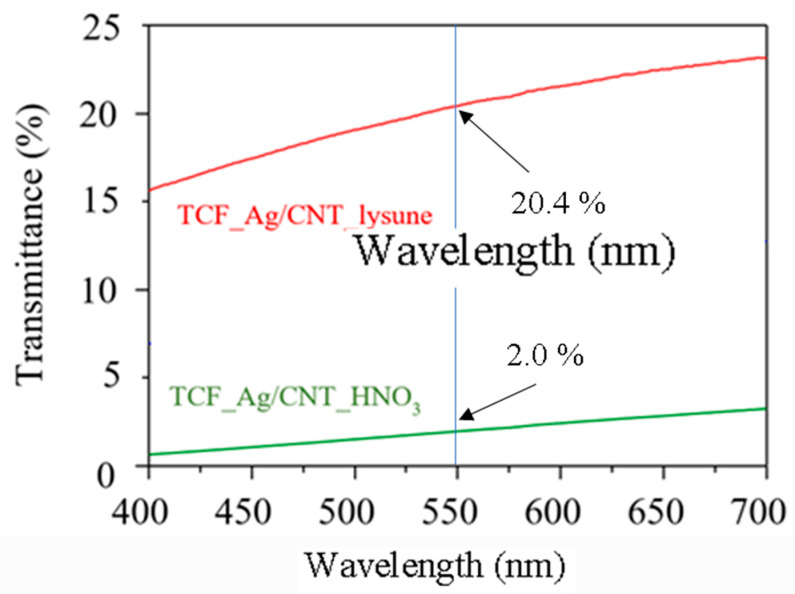
Variation of the optical transmittance of AgNP/MWCNT films with the wavelength.

**Figure 9 nanomaterials-13-01297-f009:**
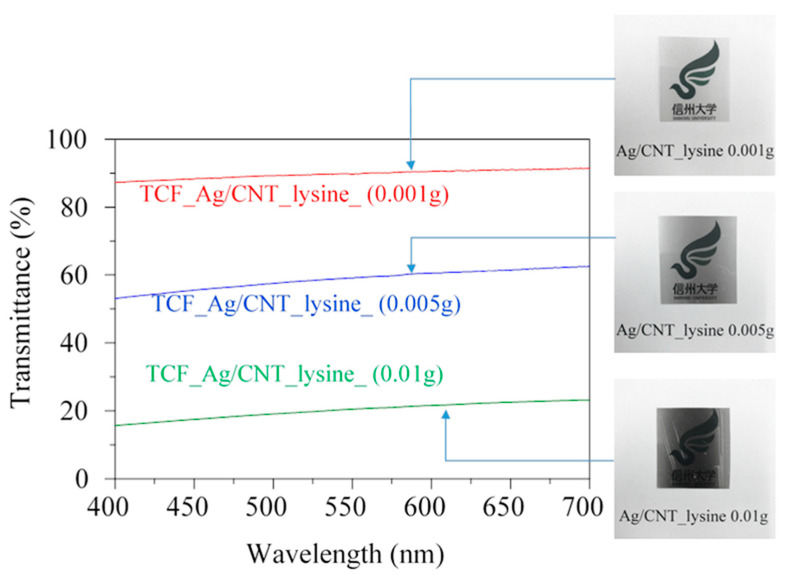
Relationship between the optical transmittance of AgNP/MWCNT films of different amounts and the wavelength.

**Figure 10 nanomaterials-13-01297-f010:**
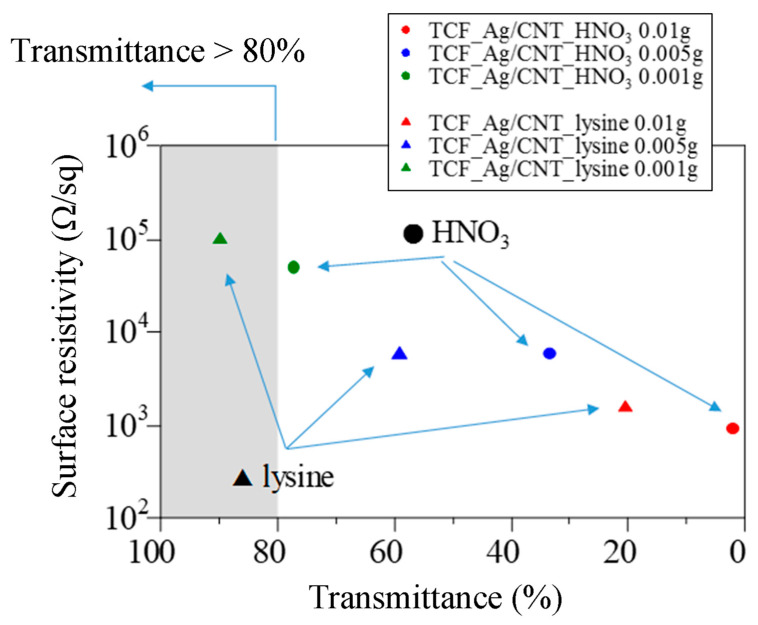
Relationship between the surface resistivity of AgNP/MWCNT films and the transmittance.

**Table 1 nanomaterials-13-01297-t001:** Ratio of Ag element (mass %) in TCF_Ag/CNT nanocomposites.

Samples	Ag
TCF_Ag/CNT_HNO_3_	84.6
TCF_Ag/CNT_lysine	76.7

## Data Availability

Not applicable.

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
