# Peer review of "Silver Nanoparticle/Carbon Nanotube Hybrid Nanocomposites: One-Step Green Synthesis, Properties, and Applications"

_nanomaterials, 2023, doi:10.3390/nano13081297_

Round 1

Reviewer 1 Report

This MS presents a simple way to get Ag nano particles on carbon nano tube surface. It gives new knowledge in this field, therefore, is worthy to be published. However, the MS need be revised for clarifying following questions.

When comparing transmittance of different samples, the same film thickness is extremely important. In this MS, I did not found thickness of all samples.

Usually, the amount of C and O are not quantitative. EDS inFig. 6 only indicated the existence of Ag. The table 1 will mislead readers.

Authors said “Almost no change of electrical resistance was observed before and after bending.” Please present the data.

Author Response

Reviewer #1:

This MS presents a simple way to get Ag nano particles on carbon nano tube surface. It gives new knowledge in this field, therefore, is worthy to be published. However, the MS need be revised for clarifying following questions.

When comparing transmittance of different samples, the same film thickness is extremely important. In this MS, I did not found thickness of all samples.

Answer:

Thanks for the reviewer's comments.

The thicknesses are less than 1 μm. We give the thickness of thin films in the revised manuscript. (see p.4)

Usually, the amount of C and O are not quantitative. EDS in Fig. 6 only indicated the existence of Ag. The table 1 will mislead readers.

Answer :

It is right. We canceled O and C in the Table 1, in the revised manuscript.

Authors said “Almost no change of electrical resistance was observed before and after bending.” Please present the data.

Thanks for the reviewer's comments.

In the revised manuscript, we presented the data as following: (see p.11)

“Moreover, we investigated the change in the electrical resistance of TCF_Ag/CNT_lysine before and after bending. The electrical resistance before bending was 22×104 Ω, and the electrical resistance after bending was 35×104 Ω.”

Reviewer 2 Report

The authors reported hybrid nanocomposites of silver nanoparticles (Ag) and multiwalled carbon nanotubes (CNTs), which successfully synthesized by a green one-step method without using any organic solvent. Two kinds of methods using nitric acid and lysine were used to modify the CNTs. There are some questions for the authors before this article can be completely accepted in the Journal. The problems and suggestions are listed as follows.

(1) The particle size distribution of the nanocomposites should be provided in the revised paper.

(2) Whether authors consider the effect of the thickness of coating layer on the properties? How to affect? That is very important for the optimal design of the membrane?

(3) As we known, the Ag nanoparticles are not stable in the atmosphere (RSC Adv., 2016, 6, 93436-93444). How about the stability of the composite?

(4) Please check the Figure 1 and Figure 9. The last step of the experimental section is not including stir. The transmittance >80% should be revised in the Figure 9.

Author Response

Reviewer #2:

The authors reported hybrid nanocomposites of silver nanoparticles (Ag) and multiwalled carbon nanotubes (CNTs), which successfully synthesized by a green one-step method without using any organic solvent. Two kinds of methods using nitric acid and lysine were used to modify the CNTs. There are some questions for the authors before this article can be completely accepted in the Journal. The problems and suggestions are listed as follows.

  • The particle size distribution of the nanocomposites should be provided in the revised paper.

Answer :

Thanks for the reviewer's comments.

We put into a TEM picture showing the distribution of AgNPs size.

  • Whether authors consider the effect of the thickness of coating layer on the properties? How to affect? That is very important for the optimal design of the membrane?

Answer :

Thanks for the reviewer's comments.

The thickness of coating layer affects the light transmittance. The transmittance decreases with increasing film thickness. In order to obtain transmittance, the film thickness of all the samples was less than 1 μm. (see p.4)

  • As we known, the Ag nanoparticles are not stable in the atmosphere (RSC Adv., 2016, 6, 93436-93444). How about the stability of the composite?

Answer :

The reduced AgNPs are covered with a reducing agent layer. Therefore, the AgNPs become very stable due to the strong binding of AgNPs with the functional groups introduced to the CNT surface through the covering layer.

(4) Please check the Figure 1 and Figure 9. The last step of the experimental section is not including stir. The transmittance >80% should be revised in the Figure 9.

Answer :

Thanks for the correct.

There was mistake on the Figures 1 and 9. We correct them in revised manuscript.
